# The Three-Level Elastoviscoplastic Model and Its Application to Describing Complex Cyclic Loading of Materials with Different Stacking Fault Energies

**DOI:** 10.3390/ma15030760

**Published:** 2022-01-19

**Authors:** Peter Valentinovich Trusov, Dmitriy Sergeevich Gribov

**Affiliations:** Department of Mathematical Modeling of Systems and Processes, Perm National Research Polytechnic University, 614990 Perm, Russia; tpv@matmod.pstu.ac.ru

**Keywords:** dislocation-based multilevel models, dislocation barriers, hardening, complex loading, cyclic deformation, microstructure evolution

## Abstract

The development of new technologies for thethermomechanical processing of metals and the improvement of the existing ones would be unattainable without the use of mathematical models. The physical and mechanical properties of alloys and the performance characteristics of the products made of these alloys are generally determined by the microstructure of materials. In real manufacturing processes, the deformation of metals and alloys occurs when they undergo complex (non-proportional) loading. Under these conditions, the formation of defect substructures, which do not happen at simple (proportional) loading, can take place. This is due to the occurrence of a great number of slip systems activated under loading along complex strain paths, which leads, for instance, to the more intense formation of barriers of different types, including barriers on split dislocations. In these processes, the formation and annihilation of dislocations proceed actively. In this paper, we present a three-level mathematical model that is based on an explicit description of the evolution dislocations density and the formation of dislocations barriers. The model is intended for the description of arbitrary complex loads with an emphasis on complex cyclic deformation.The model is composed of macrolevel (a representative macrovolume of the material that can be considered as an integration point in the finite-element modeling of real constructions), and mesolevel-1 (description of the mechanical response of a crystallite) and mesolevel-2 (description of the defect structure evolution in a crystallite) submodels. Using the model, we have performed a series of numerical experiments on simple and complex, monotonic and cyclic deformations of materials with different stacking fault energies, analyzed the evolution of defect densities, and analyzed the challenges of a relationship between the complexity of loading processes at a macrolevel and the activation of slip systems at low scale levels.

## 1. Introduction

The problems encountered in the design and manufacture of parts and structures made of metals and alloys (in most cases, polycrystals) require the development of constitutive models capable of describing changes in the physical and mechanical characteristics of materials observed over the broad range of thermomechanical treatment conditions. Material properties, such as plasticity and strength limits, corrosion resistance, electromagnetic characteristics, etc., are known to be primarily dependent on the structure of materials, and therefore, novel models should permit describing microstructure evolution at different structural-scale levels (at the level of grains, subgrains, fragments, cells, and so on). Thus, the most promising models are the models based on the introduction of internal variables ([1,2,3,4,5,6,7,8], etc.) and relying on the multilevel approach and physical theories of elastoplasticity (elastoviscoplasticity) ([9,10,11,12,13,14,15], etc.). Recent decades have seen steady progress in the development of internal variable models.

A significant amount of parts and structures used in various industries are made from metals and alloys by forming methods. The structure of polycrystalline materials in finished products is mainly determined by the deformations they experience in manufacturing processes. In real technological processes, most materials are subjected to complex loads. The formulation of constitutive relations (CRs) used to describe the properties of these materials within the framework of a phenomenological approach is associated with great difficulties [16,17,18,19]. These CRs are not universally applicable; for each material and for each loading mode, they need to be modified, and costly complex loading experiments are required. Besides, the phenomenological CRs do not allow one to describe the evolution of material structure and, consequently, they fail to predict the physical and mechanical properties and performance characteristics of finished products.

In most of the forming processes, the workpieces are subjected to loads close to the cyclic ones, including those at the stage of finishing operations. Therefore, the studies on cyclic loading, especially along complex cyclic strain paths, are of significant interest.

The results of the experimental studies on complex cyclic loading are widely discussed in the literature. Analysis of this data has revealed certain regularities [20]: a stationary mode in the curve for amplitude stresses vs. accumulated deformations at constant strain amplitudes is reached [21,22,23], and the material adapts to strain path changes, which induces additional cyclic material hardening [24,25]. The latter results in a significantly higher (tens of percent) level of steady-state amplitude stress at fixed strain amplitudes under complex loading compared to the stress amplitude that occurred under simple (proportional) loading at identical strain amplitudes [26,27,28].

Researchers note that significant restructuring of the microstructure leading to the formation of complex defect substructures can occur during non-proportional cyclic deformation [29]. Analysis of the mechanical test data and the microstructure of materials with a low stacking fault energy (SFE) has attracted special interest. It has been established that such materials are characterized by more active additional hardening, which is attributed to the formation of stable structures (barriers on split dislocations) [30,31].

Creating constitutive models intended for the description of the response of materials to cyclic complex loading is a non-trivial task; the prehistory of actions is of primary importance in the implementation of this task [32]. There are two approaches to solving this problem: (1) use of the phenomenological theories of plasticity (e.g., the endochronic theory [33,34]), or the theory of plastic flow, and formulation of a law for yield surface changes as action functionals [35,36,37], (2) use of the constitutive models relying on the explicit description of the evolution of microstructure, and definition of the mechanical properties of the material.

Some of the known macrophenomenological models can be used to describe a non-proportional cyclic deformation for specific materials and load modes with satisfactory accuracy. However, the construction of these models is a time consuming and costly procedure due to the difficulties in performing complex loading tests. One of the disadvantages of existing macrophenomenological models is their non-versatility, i.e., the necessity to “tune” them for different classes of materials and deformation processes. Besides, there arises a need for the empirical identification and verification of these models for specific alloy grades and history of changes in stress-strain state (SSS), without the possibility of applying this information to other materials.

As noted above, much attention is currently devoted to the development of an approach relying on the explicit description of the physical mechanisms of inelastic deformation and their carriers (defects of various structural-scale levels) responsible for the changes in the properties of materials. In the models of this type, several scale levels are frequently considered at once, which makes it possible to describe the evolution of the material with varying degrees of detail (depending on the formulated problem and the setup). In [10], one can find a literature review (from 1980 to 2010) and a detailed classification of models used to describe the plastic deformation processes. 

In the last few years, we have witnessed a growing interest in multilevel models that include an explicit description of the evolution of dislocations and other defect substructures [11,38]. These models [39,40,41,42,43] are based on an explicit description of the mechanisms of plastic deformation and their carriers (shears along the slip system (SS), sliding of edge dislocations, twinning, etc.), which makes it possible, e.g., to derive physically substantiated hardening laws [13,44,45,46,47].

The models of this class have wider applicability than the macrophenomenological models because the main mechanisms of deformation and evolution of the microstructure are practically the same for a wide range of materials and effects. However, the implementation of these models requires the identification of a great number of parameters entering the relations for describing the structure evolution processes at various structural-scale levels. This problem can be solved using the submodels of deeper scale levels, for example, by applyingthe dislocation or molecular dynamics methods [45,48,49].

The work [50] presents the results of an experimental investigation and numerical modeling of aluminum samples loading along various deformation trajectories. To describe plastic strain rate, the model uses the classical Hutchinson equation with the introduction of a matrix to describe non-isotropic hardening. The results of applying the model are in good agreement with the experimental data. The clear advantages of the work include the combination of natural and numerical experiments within one work, which gives researchers the opportunity to make plans for full-scale experiments for the purpose of identifying and verifying the parameters of a mathematical model. The disadvantages of this model include implicit accounting for the evolution of the mechanism of plastic deformation—the movement of dislocations; the relationships are written in phenomenological (for the level of the crystal’s RV) form.

The results of using a two-level model of crystal plasticity FEM to study the level of stored energy during plastic deformation depending on the initial texture are contained in [51]. At each FE integration point, the plastic strain rates are determined by averaging over 400 grains, for a set of grains Voigt’s hypothesis is accepted at the points of integration. In each grain the plastic component of the displacement velocity gradient is set according to the shear rates along the SS; for the shear rates, the viscoplastic law is used, which takes into account the deformation temperature. The hardening law takes into account temperature dependence and latent hardening. The model under consideration is applied to analyze uniaxial monotonic (tension) and cyclic (along a triangle) loading of specimens made of monocrystalline and polycrystalline austenitic steel. A significant effect of texture on the stored energy level is shown; its greatest value is achieved in a single-crystal sample. Comparison of the nature of cyclic loading deformation under isothermal and adiabatic conditions indicates a significant difference in the mechanical behavior of the samples. In the case of cyclic deformation, it is shown that the change in the stored energy is cyclical.

A modification of an elastic-viscoplastic model of the CPFEM type focused on describing the behavior of a two-phase titanium alloy (HCP lattice) during thermomechanical processing, is considered in [52]. The hardening law is based on evolutionary equations for the dislocation density; in this case, along with edge dislocations, screw dislocations are also included. Particular attention is devoted to the description of the dynamic recrystallization process, the main driving force of which is assumed to be an excessive accumulation of dislocations. 

A dislocation density based crystal plasticity model was presented in [53]. The developed model is able to describe cyclic softening in steel alloy. The model’s advantage is the description of the interaction between the precipitates and the dislocations. The slipping rates are described using Hutchinson’s equation. Evolutionary equations for dislocation densities on slip systems include taking into account the formation and annihilation of dislocations. Model predictions of macroscopic behavior are shown to be in good agreement with experimental data. In [54] was presented crystal-plasticity approach, based on the dislocation density model. The model includes three different types of dislocations: forward, reverse and named by authors latent dislocations (that are inactive at the moment), which are selectively activated depending on load path changes. Of particular interest is the form of the hardening law, which takes into account the three introduced types of dislocations. The procedure for identifying model parameters is presented. It is shown that the results of numerical simulation are in satisfactory agreement with the data of natural mechanical experiments for various loads.

Summarizing the analyzed sources, it can be noted that in the works known to the authors (including those mentioned above), insufficient attention is paid to processes of dislocations splitting, creating of sustainable barriers (Lomer-Cottrell and Hirth locks), and a description of the effect of the non-proportional loading on the SS hardening. We can also note little attention to the formation of dislocations due to the work of Frank–Read sources.

Analysis of existing approaches to describing complex cyclic deformation has indicated the feasibility of constructing a model relying on a direct description of crystal lattice defects and their evolution and including the formulation of physically substantiated hardening laws. In order to develop the model, it is necessary to take into account the effect of the stacking fault energy (SFE) value on the evolution of microstructure, especially, on the formation of barriers on split dislocations and their effect on hardening.

In this work, we construct constitutive models within the framework of the above-described approach, which is based on the introduction of internal variables (IVs). The applied internal variables are considered as explicit IVEs and implicit IVIs; IVEs are directly part of the structure of CRs at this scale level; IVIs contain variables that describe the microstructure of deeper structural-scale levels and are included as variables in the evolutionary equations at a given scale level. The same IVs, yet at different structural levels, act as both IVEs and IVIs.

## 2. The Three-Level Model for Describing Complex Cyclic Deformation

The model developed in this study belongs to the class of three-level models (Figure 1). We introduce the following structural-scale levels to describe the deformation of a polycrystalline sample: macrolevel, mesolevel-1, and mesolevel-2. We assume that the key mechanism governing the inelastic deformation of the sample is the motion of edge (complete and split) dislocations. At the mesolevel-2, this mechanism is described relative to the introduction of dislocation densities on slip systems (SS) and their velocities. At mesolevel-1, the inelastic deformation is considered in terms of shear rates along the crystallographic SS, which are determined by the Orowan equation on the basis of the parameters determined at mesolevel-2. To connect the levels, the explicit and implicit internal variables (IVs) [55] determined from the closing equations are introduced into the structure of constitutive relations at every scale level. It is worth noting that these IVs are capable of describing the deformation processes at deeper scale levels compared to the one considered here.

A representative macrovolume is hierarchically senior with respect to a set of mesolevel-1 elements. At the macrolevel, the behavior of the representative volume (RV) of the materialis described in terms of continual macrovariables, including the elastic and plastic constituents of the stretching tensor, the stress tensor and the rate of its change. The macro-level elements can be used to determine the response of the material at integration points when the finite element method is applied to solve the boundary value problems. At the macrolevel, the loading regime is set according to the law of a prescribed change in the velocity gradient with time.

Under the Voigt (Taylor) hypothesis, the kinematic variables are transmitted from the macrolevel to the mesolevel-1 as an action. At mesolevel-1, a consideration is carried out in terms of shear stresses and shear rates along slip systems. The shear stresses on slip systems determined at the mesolevel-1 are transferred as an effect to the mesolevel-2, where the evolution of defect densities is studied, and the hardening along slip systems is described. The shear rates along the SS, determined using Orowan’s equation, are transferred from the mesolevel-2 to the mesolevel-1.

The mesoscale-2 submodel uses the division of dislocation densities by slip systems into positive and negative, depending on the direction of an extra plane. The variables in the mesolevel-2 submodel are as follows: dislocation densities (positive and negative), densities of dislocation sources on the SS, and the sliding barriers formed. In constructing the model, we have paid particular attention to a physically substantiated description of dislocation sliding and dislocation interactions, including those of split dislocations accompanied by the formation of Lomer–Cottrell and Hirt locks.

Hereinafter, the “related” characteristics of the macrolevel and the meso-level-1 are denoted by the same letters (the macrolevel—by uppercase letters, and the mesoscale—by lowercase letters). The kinematic effect specified at the macrolevel $Z=\hat{\nabla }V^{T}$ is transmitted to the mesolevel-1 using the Voigt (Taylor) hypothesis:(1)$$z\left(t\right)=Z\left(t\right)=\hat{\nabla }V^{T}$$
where $\hat{\nabla }$ is the gradient (Hamilton) operator defined in the actual configuration, and **z** is the velocity gradient (strain rate change) at the mesolevel-1.

At the mesolevel-1, we apply the plastic constituent of the strain rate **z***^in^*, which is calculated from the dislocation velocities and densities found in the mesolevel-2 submodel; the components of the Cauchy stress tensor are also defined. The stress tensor **σ** is averaged over the aggregate of crystallites (the macrolevel RV), which yields the Cauchy macrostress tensor **Σ**. Evaluation of the stresses **σ** and unit vectors of the normal and the direction of sliding provides the shear stresses $\tau^{\left(k\right)}$ acting on each *k*-th SS at the mesolevel-1. The shear rates ${\dot{\gamma }}^{\left(k\right)}$ are determined at the mesolevel-2 from dislocation velocities and dislocation current densities. They are used to calculate the inelastic strain rate at the mesolevel-1. To describe the evolution of the internal variables of the mesolevel-1, we apply the following system of equations (the numbers of crystallites are omitted here and below) [15]:(2)$$\left\{ \begin{array}{l} \dot{\sigma }+\sigma \cdot \omega -\omega \cdot \sigma =п:(z-\omega -z^{in}),\\ z=z^{e}+z^{in},\\ \omega =\dot{o}\;\cdot \;o^{T},\\ z^{in}=\sum_{k=1}^{n}{\dot{\gamma }}^{\left(k\right)}b^{\left(k\right)}n^{\left(k\right)},\\ \tau^{\left(k\right)}=b^{\left(k\right)}n^{\left(k\right)}:\sigma , \end{array}\right..$$
where operation “·” is scalar production, “:” is the double dot product, the symbol of tensor multiplication (between vectors and tensors) and scalar (between scalars, scalar and tensor) is not indicated, **п** is a 4-valent elastic tensor, **z***^e^* and $z^{in}$ are the elastic and inelastic constituents of the strain rate, o is the orientation tensor (orthogonal tensor transforming the basis of the laboratory coordinate system into the basis of the crystallographic coordinate system), **ω** is the spin of the corotational coordinate system related to the lattice [56], **b** is the unit vector in the direction of the Burgers edge dislocation vector, **n** is the unit normal vector to the plane of dislocation sliding, ${\dot{\gamma }}^{\left(k\right)}$ is the shear rate along the slip plane, and *k* is the number of the slip system (SS). The evolution of the dislocation micro (sub)structure is described at the mesolevel-2.

The shear stresses $\tau^{\left(k\right)}$ acting along SS are transmitted from the mesolevel-1 to the mesdolevel-2. In order to describe the evolution of defect densities on the SS, we take into account the most important mechanisms: dislocation nucleation due to the operation of Frank–Read sources, the annihilation of dislocations of different signs on one SS, and changes in barrier densities on the SS.

At the mesolevel-2, we calculate the average velocities of positive and negative dislocations $V_{\pm }^{\left(k\right)}$, which depend on the value of tangential stresses $\tau^{\left(k\right)}$, temperature *θ*, densities of positive and negative dislocations $\rho_{+}^{\left(k\right)},\rho_{-}^{\left(k\right)}$ and the barrier density $\rho_{bar}^{\left(k\right)}$ on the SS. To this end, we apply an additive decomposition of the critical shear rate ${\dot{\tau }}_{c}^{\left(k\right)}$ into constituents that depend on the changes in the dislocation densities along the SS ${\dot{\tau }}_{c\_dis}^{\left(k\right)}$ and on the changes in barrier densities ${\dot{\tau }}_{c\_bar}^{\left(k\right)}$. Shear rates are found using Orowan’s velocity form equation. The crystallite behavior at the mesolevel-2 is described by the following system of equations (in general form):(3)$$\left\{ \begin{array}{l} V_{\pm }^{\left(k\right)}=f_{1}(\tau^{\left(k\right)},\tau_{c\pm }^{\left(k\right)},\theta )sign(\tau^{\left(k\right)}),\\ {\dot{\gamma }}^{\left(k\right)}=(\rho_{+}^{\left(k\right)}V_{+}^{\left(k\right)}-\rho_{-}^{\left(k\right)}V_{-}^{\left(k\right)}){\left|b\right|}_{}^{\left(k\right)},\\ {\dot{\rho }}_{\pm }^{\left(k\right)}=f_{2}(\tau^{\left(k\right)},\tau_{c\pm }^{\left(k\right)},\theta ,\rho_{\pm }^{\left(k\right)},\rho_{bar}^{\left(k\right)}),\\ {\dot{\rho }}_{bar}^{\left(kl\right)}=f_{3}(\tau^{\left(k\right)},\tau_{c\pm }^{\left(k\right)},\theta ,\rho_{\pm }^{\left(k\right)},\rho_{bar}^{\left(k\right)}),\\ \tau_{c0\pm }^{\left(k\right)}=\tau_{c\_lat\pm }^{\left(k\right)},\\ {\dot{\tau }}_{c\pm }^{\left(k\right)}={\dot{\tau }}_{c\_dis\pm }^{\left(k\right)}+{\dot{\tau }}_{c\_bar\pm }^{\left(k\right)}, \end{array}\right.$$
where θ is the temperature.

The average dislocation velocities ($V_{+}^{\left(k\right)},V_{-}^{\left(k\right)}$) play a significant role in describing dislocation reactions. Note that the velocities of the dislocations of opposite signs on the same crystallographic plane can differ not only in sign but in magnitude as well, in this case, the densities of dislocations of different signs can also differ. For dislocations of different signs, the evolving critical shear stresses near one-sided barriers, such as grain boundaries and rigid inclusions differ, and as a result, both the densities and velocities of dislocation motion differ. We also should take into account the fact that the motions of dislocations of both signs along the same SS contribute to shear rate along this system, which is why the Orowan equation [57] in rate form is slightly modified:(4)$${\dot{\gamma }}^{\left(k\right)}=(\rho_{+}^{\left(k\right)}V_{+}^{\left(k\right)}-\rho_{-}^{\left(k\right)}V_{-}^{\left(k\right)}){\left|b\right|}_{}^{\left(k\right)}.$$

The projections of the dislocations velocities on SS are determined as algebraic values that are calculated for local SS coordinate systems. This procedure is justified by the need to describe dislocations annihilation, where the arithmetic values of these velocities must add up:(5)$$V_{\pm }^{\left(k\right)}=b_{\pm }^{\left(k\right)}\cdot V_{\pm }^{\left(k\right)}.\;$$

The mean dislocation velocities are defined by the following relations [58]: (6)$$\left\{ \begin{array}{l} V_{+}^{\left(k\right)}=l_{+}^{k}v\exp(-\Delta G_{*}^{k}/k_{B}\theta )\;H\left(\left|\tau^{\left(k\right)}\right|–\tau_{c+}^{\left(k\right)}\right)\;sign(\tau^{\left(k\right)}),\\ V_{-}^{\left(k\right)}=-l_{-}^{k}v\exp(-\Delta G_{*}^{k}/k_{B}\theta )\;H\left(\left|\tau^{\left(k\right)}\right|–\tau_{c-}^{\left(k\right)}\right)\;sign(\tau^{\left(k\right)}), \end{array}\right.$$
where $\Delta G_{*}^{k}$ is the activation energy of dislocation motion (dependent on the properties of the lattice, that also include critical stress $\tau_{c\pm }^{\left(k\right)}$ and the shear stress $\tau^{\left(k\right)}$ on the SS), $k_{B}$ is Boltzmann constant, θ is the temperature, H is the Heaviside function, *l^k^* is the average length of the mean free path of the dislocations on the *k*-th SS, and v is the Debye frequency.

During the isothermal plastic deformation, the density of dislocations on the SS increases. The Frank–Read sources which produce closed expanding dislocation loops are identified as intragranular dislocation sources. Experimental studies provided evidence that a single source is able to generate a limited number of loops. To describe the generation of dislocations, we introduce here the densities of Frank–Read sources $\rho_{src}^{\left(k\right)}$ (dimension m^–3^); changes in these densities are expressed by the following relation: (7)$$\begin{array}{l} {\dot{\rho }}_{src}^{\left(k\right)}=\sum_{j}L\rho_{bar}^{kj}\rho^{j}[l^{j}v\exp(-\Delta G_{*\pm }^{j}/k_{B}\theta )\;H\left(\left|\tau^{\left(j\right)}\right|–\tau_{c\pm }^{\left(j\right)}\right)],\\ \rho_{0src}^{\left(k\right)}=\rho_{0src}, \end{array}$$
where $\rho_{bar}^{kj}$ is the density of the barriers at the intersection of *k* and *j* SS (it hinders or prevents the dislocation motion on both SS), *L* is the distance between the barriers in the source. Source contribution to the growth of dislocation densities is proportional to the current density of the loops and is nonzero only when the acting tangential stresses are higher than the critical stresses $\tau_{src}^{}$ [59]:(8)$$\begin{array}{l} {\dot{\rho }}_{nuc}^{\left(k\right)}=r_{av}\rho_{src}^{\left(k\right)}v{\langle \frac{\left|\tau^{\left(k\right)}\right|}{\tau_{src}}-1\rangle }^{p},\\ \tau_{src}=\frac{A\mu b}{2\pi L}\left(\ln\frac{L}{r_{0}}+B\right), \end{array}$$
where $r_{av}$ is the mean radius of the loop, v is the Debye frequency, *A*, *B*, are the dimensionless material parameters, *μ* is the shear modulus, $r_{0}$ is the minimum length of the fastened segment capable of generating dislocation loops, *b* is the Burgers vector value, and 〈⋅〉 are the Macaulay brackets (〈x〉=x⋅H(x)).

Dislocations annihilation occurs when dislocations of different signs meet each other on the same SS. This reaction is possible if two dislocations of opposite signs are at a short distance from each other on the adjacent SSs. Dislocation annihilation is observed most frequently during the reverse loading experiments. The dislocations of opposite signs, which occur on the same slip system, attract each other, and those on parallel slip systems can climb towards each other, annihilating when approaching the distance *h_ann_*. The number of the reacted dislocations per unit time is proportional to the swept volume and the dislocation density on slip systems. In order to describe this annihilation, we use the following relation [60]:(9)$${\dot{\rho }}_{\pm }^{(k)ann}=-h_{ann}^{}\rho_{+}^{\left(k\right)}\rho_{-}^{\left(k\right)}\left|V_{+}^{\left(k\right)}-V_{-}^{\left(k\right)}\right|.$$

The generation and evolution of barriers are described using the constructed barrier density matrix $R_{\mathrm{bar}}^{kl}$. In this matrix, the unit values at the intersection of row *k* and column *l* denote the number of the SS in which the split dislocations occur, reacting upon crossing to form the Lomer–Cottrell and/or Hirt barriers. The value of the SFE under any loading affects the critical stresses and hardening; for example, in materials with a low SFE, dislocations tend to split, which makes it difficult for their non-conservative movement (cross-slips) and overcoming barriers. Split dislocations tend to form rigid barriers, especially under complex cyclic loading. Other components of the matrix $R_{\mathrm{bar}}^{kl}$ are zero. The rate of change in the density of barriers depends on the density of dislocations on the reacting systems, the acting shear stresses and temperature [61]:(10)$$\begin{array}{l} {\dot{\rho }}_{bar\pm }^{kl}=x_{d}R_{\mathrm{bar}}^{kl}\rho_{\pm }^{l}\rho_{\pm }^{k}[l^{k}v\exp(-\Delta G_{*\pm }^{k}/k_{B}\theta )],\\ x_{d}=\frac{b}{8\pi \varepsilon_{SFE}}=\frac{b^{2}G}{8\pi \gamma_{SFE}}, \end{array}$$
where α is the dimensionless parameter, $x_{d}$ is the mean dislocation splitting width, $\varepsilon_{SFE}$ is the dimensionless value of SFE (εSFE=γSFEGb, $\gamma_{SFE}$—stacking fault energy). The matrix of the density of barriers on the SS contains information about all 24 barriers, which can form on split dislocation. The components of the matrix ${\dot{\rho }}_{bar}^{kl}$ describe the rate of change in the density of barriers formed at the intersection of dislocations on the *k*-th and *l*-th SSs. Thus, we can calculate the densities of the barriers that occurred in the course of the reactions described by the matrix $R_{\mathrm{bar}}^{kl}$, which allows describing the interaction of dislocations of both signs.

In formulating a hardening law, we have made a hypothesis that supports the possibility of an additive decomposition of the critical stresses of the SS into contributions from the lattice resistance (a value that depends only on temperature), from the dislocations stress fields and from the barriers formed on split dislocations. The matrix components describing the effect of accumulated defects were determined by making an estimate of the influence of dislocations of different SSs on each other.

The well-known solution for a single dislocation in an isotropic elastic medium was used to evaluate how the stress fields of one dislocation affect another dislocation. Based on this solution, we constructed a matrix *M^ki^* for making estimates of the levels of interactions between the *k*-th and *i*-th SSs at the intersection of the *k*-th row and *i*-th column of the matrix *M^ki^*.

The introduction of a split dislocation barrier as a complex of two partial and sessile dislocations enables assessing the effect of the barrier on the strengthening of the *k*th SS due to the barrier on the *i*-th SS. To describe this effect, we applied the matrix *B^ki^*, which was constructed like the matrix *M^ki^*. The matrix components were normalized to the initial critical stresses of the corresponding slip systems (Peierls stresses) $\tau_{c0\pm }^{\left(k\right)}=\tau_{c\_lat\pm }^{\left(k\right)}$ (they are dimensionless quantities). Using the introduced matrices, the evolutionary relations showing how the critical shear stress and its components change can be written as [62]:(11)$$\begin{array}{l} \tau_{c0}^{\left(k\right)}=\tau_{c\_lat}^{\left(k\right)},\\ {\dot{\tau }}_{c\pm }^{\left(k\right)}={\dot{\tau }}_{c\_dis\pm }^{\left(k\right)}+{\dot{\tau }}_{c\_bar\pm }^{\left(k\right)},\\ {\dot{\tau }}_{c\_dis\pm }^{\left(k\right)}=\alpha b\tau_{c0\pm }^{\left(k\right)}\sum_{i=1}^{n}\frac{M^{ki}}{2\sqrt{\rho_{\pm }^{i}}}{\dot{\rho }}_{\pm }^{i},\\ {\dot{\tau }}_{c\_bar\pm }^{\left(k\right)}=\beta b\tau_{c0\pm }^{\left(k\right)}\sum_{i=1}^{n}\frac{B^{ki}}{2\sqrt{\rho_{bar\pm }^{ki}}}{\dot{\rho }}_{bar\pm }^{ki}, \end{array}$$
where α,β are the dimensionless material parameters.

In view of the above, we provide a brief description of the general structure of the model, levels and the relationships between them. Each macrolevel element includes a finite set (from several hundreds to thousands) of mesolevel-1 elements having different lattice orientations with respect to the characteristic loading axes, as well as to the axes of the laboratory coordinate system; each mesolevel-1 element coincides in scale with one of the elements of the mesolevel-2.

Due to the problem of nonlinearity, we apply a step-by-step algorithm to find a solution to this problem. Let us briefly consider the time-stepping algorithm intended for the assessment of changes in the state of the material. At the macrolevel, we set kinematic actions and apply the Voigt (Taylor) hypothesis on the equality of velocity gradients to establish a relationship between the macrolevel and the mesolevel-1 submodels. The elements of mesolevel-1 and mesolevel-2 are connected from top to bottom by transferring the acting shear stresses along slip systems to the current step.

At the mesolevel-2, the rates of change of IVIs and IVEs are evaluated, and the average shear rates along the SS are transmitted as a response to the mesolevel-1, where the inelastic component of the strain rate and the rate of change of the Cauchy stress tensor is determined. By averaging the parameters determined in the mesolevel-1 elements, at the macrolevel (as at the level of the representative macrovolume of the material), the macrostresses are established at the end of the time step. At the macrolevel, new kinematic actions are set, and the time step is repeated (Figure 2).

## 3. The Results of Application of the Model to Describing Cyclic Deformation

The model presented above was used to describe the response of an fcc polycrystal. The RV of the macrolevel is assumed to consist of 125 crystallites (grains); distribution of orientations according to a uniform law in the reference configuration is adopted. The numerical experiments were carried out on the model copper and brass RVs, which have different values of elastic tensor components, lattice initial resistance, SFE magnitude, and hardening law parameters. The initial dislocation densities are assumed to be the same and equal to 10^9^ m^–2^, the initial barrier densities are equal to zero.

Determining the parameters of a model is a separate complex task, for the solution of which several approaches are used. Some of the parameters that can be determined correspond to the values known from physical theories. The second part of the parameters, for example, the components of matrices for describing hardening is determined by the description of dislocations as ideal objects in an isotropic elastic medium. The third part of the parameters, which are key, is determined using direct search methods, by solving the optimization problem by the Nelder–Mead method, which implies a large number of numerical experiments. All important parameters of the model are presented in Table 1.

In the numerical experiments, we consider a kinematically-driven loading. The simple and complex loading experiments are carried out under the combined action of tension and shear strain, and the deformations are limited to maximum strain amplitudes and are assumed to be the same for the materials with different SFE. The main purpose of our study is to analyze the difference in the response of materials to loads of different complexity, which manifests itself even at the early stages of inelastic deformation. At small displacement gradients, the crystallite lattice rotations can be neglected, and the loading mode can be prescribed in terms of small deformations and their rates. Kinematic actions are specified by the velocity gradient components in the framework of the laboratory coordinate system:(12)$$\left(\nabla v^{T}\right)=\left(\begin{array}{ccc} d_{11} & d_{12} & d_{13}\\ d_{21} & d_{22} & d_{23}\\ d_{31} & d_{32} & d_{33} \end{array}\right).$$

Figure 3 presents the results of numerical simulations for the copper and brass RVs subjected to simple (proportional) cyclic loading; it should be noted that the loading diagrams for shear and tensile loads are practically indistinguishable. During the simple cyclic deformation tests, we faced the experimentally-proved fact that the stresses amplitudes reach stationary values at limited deformation amplitudes (3%): the maximum stresses were 41.6 MPa for the copper sample subjected to shear loading, and the maximum amplitude stresses were 50.3 MPa for the brass sample subjected to tension-compression loading.

The developed model was applied to describe the complex cyclic deformation by taking into account different degrees of non-proportionality (the out-of-phase changes of the strain tensor components). The loading program was specified by the following relationships:(13)d12=dsmsin(ωt+ϕ),d11=drmsin(ωt),Pm=dsm3drm,d22=d33=–12d11,d23=d31=0,
where $d_{\mathrm{rm}}$, $d_{\mathrm{sm}}$ are the amplitude values of longitudinal and shear deformations, ϕ is the angle of mismatch between tension-compression and shear deformations, $P_{m}$ is the coefficient of proportionality of the deformation modes, and ω is the cyclic deformation frequency. In the strain space, the loading trajectory is an ellipse, the elongation along the main axes characterizes the ratio of amplitudes for tension and shear, and the angle of the ellipse to its major axes is determined by the value of the mismatch angle. All trajectories considered in the two-dimensional strain space are located inside a circle of the prescribed radius of the amplitude of strain intensity, and they have at least two points of contact with it. Figure 4 presents the calculation results for the copper and brass RVs subjected to loading realized according to Equation (12). The use of the approach with the introduction of the internal variables makes it possible to determine the density of dislocations and barriers in average and in each crystallite, during the loading process (Figure 5). We can see a quantitative difference in the formation of barriers and the nucleation of new dislocations in copper and brass experiments. The main difference in the parameters used is the value of the stacking fault energy; in brass, the formation of barriers occurs more intensively.

Of particular interest is the description of the phenomenon of additional hardening, that is, an increase in the amplitude of stress intensity under complex loading as compared to simple cyclic loading at the same amplitudes of strain intensity. During the numerical experiment, taking into account the given kinematic action, the maximum stress intensities were determined (after reaching the stationary values of the curve, the stress amplitude—the accumulated strain). To estimate the amount of additional hardening in terms of the obtained maximum stress intensities, the maximum stress intensities found in the simple cyclic loading experiments were subtracted and referred to them, which yielded the percentage value of additional hardening to the reference additional hardening $\Delta_{ad}$. By varying the value of the mismatch angle and the coefficient of non-proportionality, we determined the number of additional hardening values; the calculation results are summarized in Table 2.

Analysis of the results of numerical experiments shows how the amount of additional hardening depends on the complexity of loading, it is shown in Figure 6.

The obtained dependencies are in qualitative agreement with the results of the full-scale experiments. The evolution of the density of defects and barriers formed on split dislocations was analyzed in a separate experiment. We note that, during the deformation of the brass sample with a relatively low SFE, the barriers on split dislocations are formed more intensively, which leads to a greater (in comparison with copper) additional hardening.

To analyze the dependence of the amount of additional cyclic hardening $\Delta_{ad}$ on the orientation of the initial part of the deformation trajectory in the deformation space, we have performed a series of numerical experiments on the closed trajectories having the form of a square (Figure 7, schemes (a) and (b). For the polycrystalline brass samples, the value $\Delta_{ad}$ ranged from 8.2% (trajectory (b)) to 8.6% (trajectory (a)), depending on the way used to reach the closed cycle. It should be noted that the difference in the magnitude $\Delta_{ad}$ can be attributed to the differences in the initial evolution of dislocation densities: under simultaneous tension and shear, the densities of dislocation and barriers increase significantly.

## 4. Conclusions

The three-level dislocation-oriented model was applied to describe the deformation of polycrystalline samples under a wide range of simple and complex kinematic loads. An emphasis was paid to the evolution of defect densities and the formation of barriers on split dislocations. The model is based on the explicit description of the behavior of complete and split dislocation aggregates on slip systems, which permits one to trace in detail the mechanisms of deformation and hardening of materials along the macroscopic strain paths of arbitrary complexity.

In order to illustrate the validity of the model, we used it to describe the processes of simple and complex, monotonic and cyclic deformation of fcc polycrystalline samples with significantly different stacking fault energies. It is shown that the materials with low stacking fault energies subjected to complex cyclic loading exhibit more intense additional cyclic hardening compared to the materials with relatively high stacking fault energies. An analysis of the RV dislocations and defects densities, which allows the model, showed a qualitative agreement: in a material with a lower SFE (brass), barriers are formed with greater intensity, which also leads to a more active formation of sources of dislocations and production of new dislocations. The data obtained indicate that the model can qualitatively describe the evolution of dislocation densities; however, the development of this model requires further improvement of the relationships for the evolution of defect densities and their parameters.The results of numerical experiments obtained using the model proposed are in satisfactory qualitative agreement with the known experimental data.

## Figures and Tables

**Figure 1 materials-15-00760-f001:**
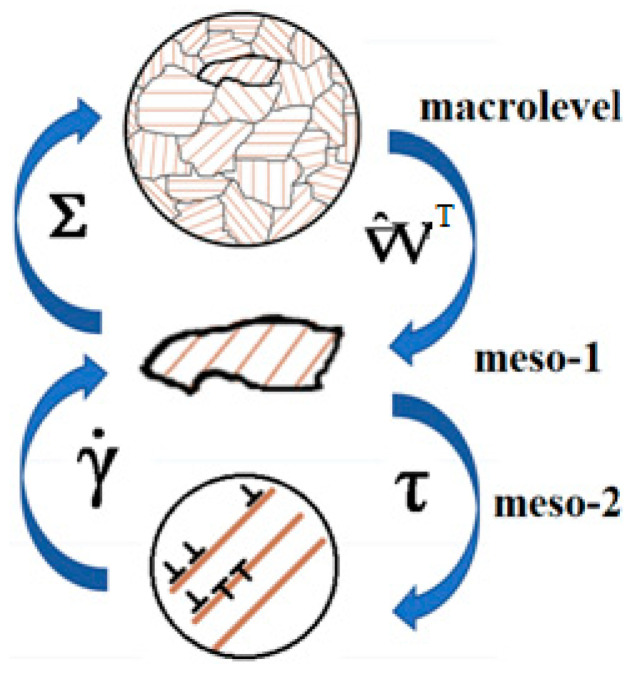
Schematic representation of the model (transfer of influences and determination of response).

**Figure 2 materials-15-00760-f002:**
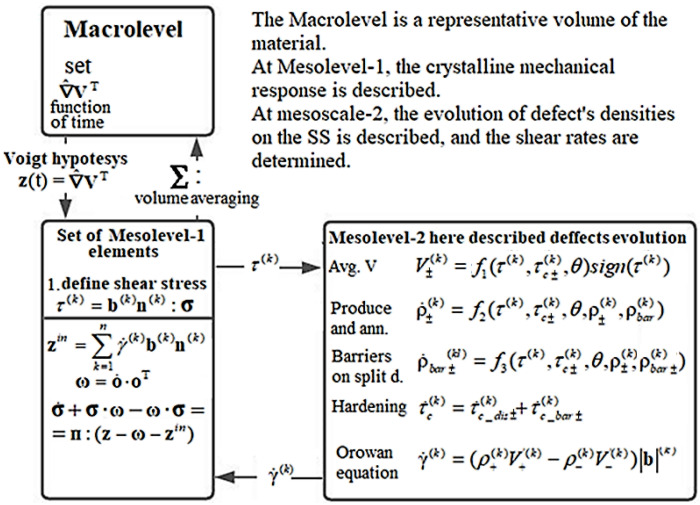
Schematic flow chart of model’s algorithm.

**Figure 3 materials-15-00760-f003:**
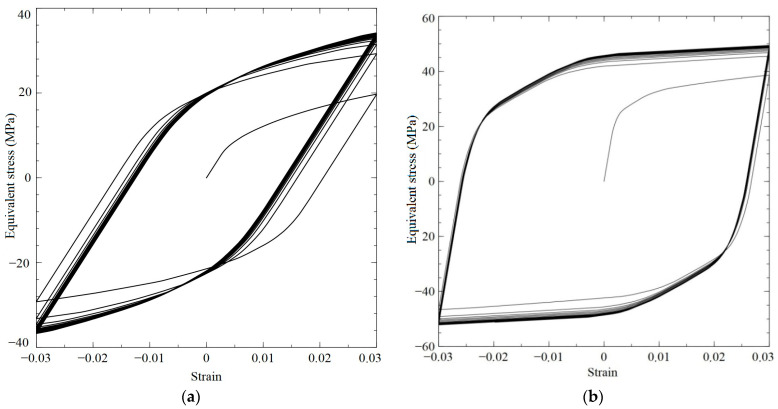
Relationship between stress and strain intensities obtained in the numerical simulations of simple cyclic loading experiments on the copper macrosample subjected to tension-compression deformation (**a**) and on the brass polycrystal sample under tension-compression deformation (**b**); maximum strain intensity −3%, 125 crystallites.

**Figure 4 materials-15-00760-f004:**
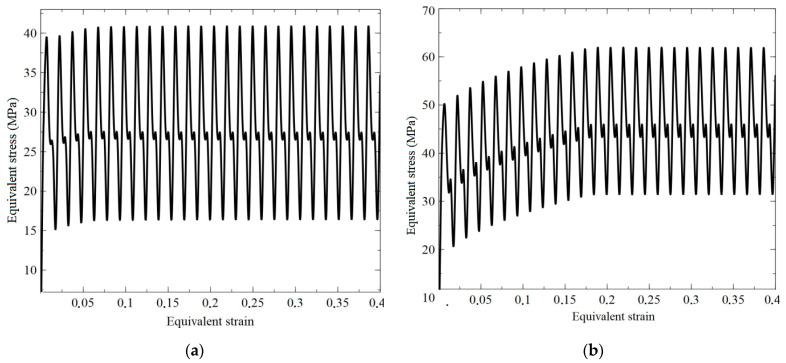
Dependence of stress intensity on accumulated strain intensity at $P_{m}$ = 1, maximum strain amplitude—1.5%, (**a**)—copper, (**b**)—brass, ϕ = 60°, 125 crystallites.

**Figure 5 materials-15-00760-f005:**
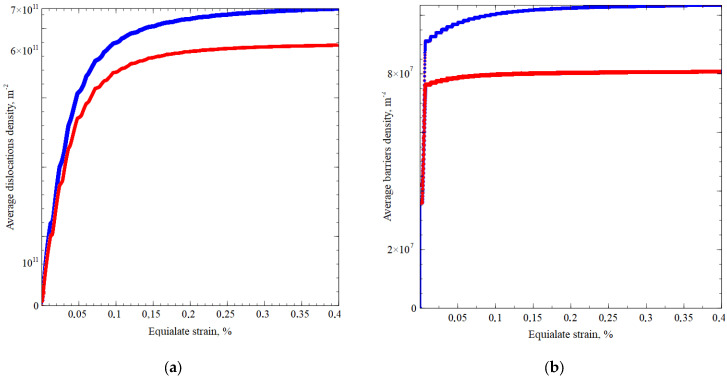
Dependence of average dislocations density (**a**), copper (red), brass (blue), and dependence of average barriers density (**b**), copper (red), brass (blue), at $P_{m}$ = 1, ϕ = 60°, maximum strain amplitude—1.5%, 125 crystallites.

**Figure 6 materials-15-00760-f006:**
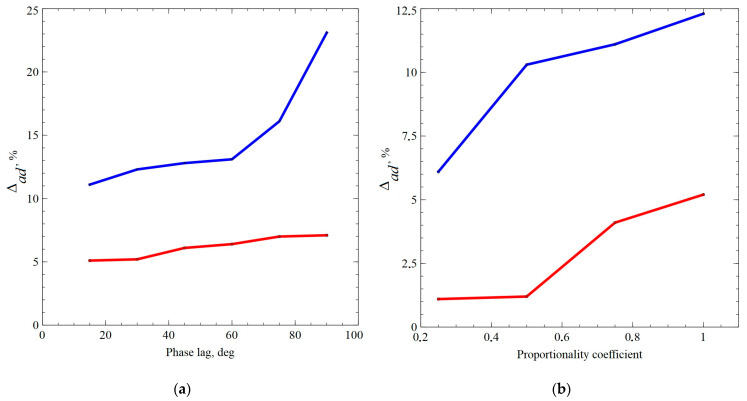
Dependence of the amount of additional hardening on the degree of non-proportionality of cyclic loading for the angle lag (**a**), and the change in the ratio of shear amplitude to tension (**b**) (brass—blue graph, copper—red graph).

**Figure 7 materials-15-00760-f007:**
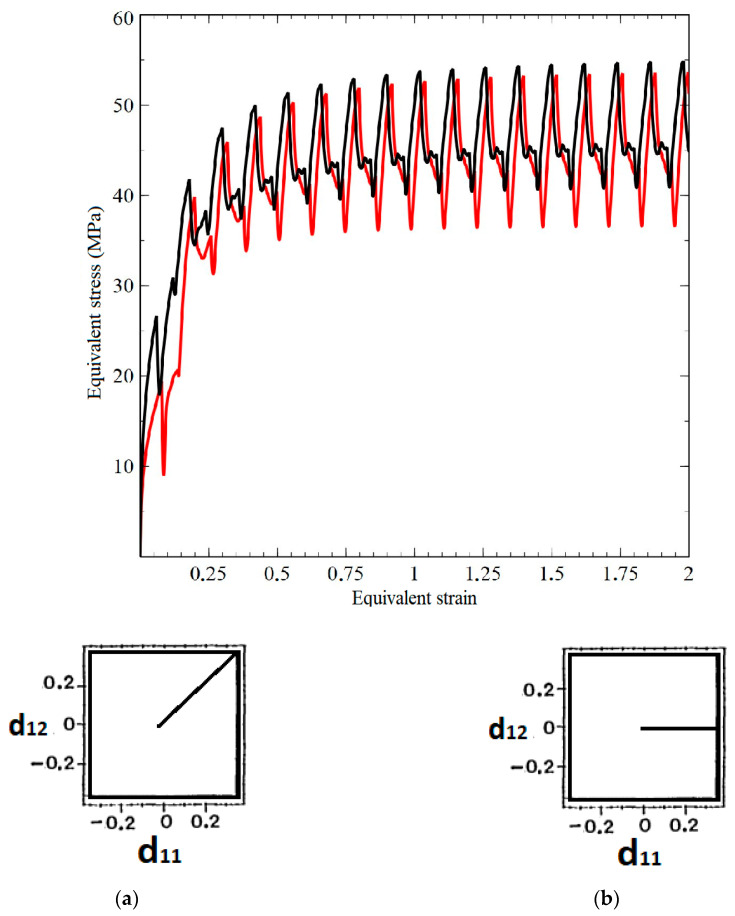
Dependence of maximum effective stresses on accumulated strain intensity in the complex cyclic loading experiments on the polycrystalline bras sample loaded along two trajectories (**a**)—black, (**b**)—red graph.

**Table 1 materials-15-00760-t001:** Model parameters, source, meaning and equation.

Variable	Magnitudes	No. Equation	Source
$\rho_{+}^{\left(k\right)},\rho_{-}^{\left(k\right)}$ at t = 0	10^9^ m^−2^	4	[59]
$\rho_{bar}^{\left(k\right)}$ at t = 0	0 m^−2^	10	hypothesis
b (for copper and brass similarly)	0.2556 × 10^−9^ m	4	[59]
v	7.11 × 10^12^ s^−1^	6	[61]
$\Delta G_{*}^{k}$	0.206 eV	6	[19]
L	5 b	7	numerical identification
$r_{av}$	15 b	8	hypothesis
$r_{0}$	3 b	8	numerical identification
A	139	8	[59]
B	5	8	[59]
$h_{ann}^{}$	5 b	9	hypothesis
$R_{\mathrm{bar}}^{kl}$	Matrix, 24 × 24 elements, dimensionless	10	lattice’s characteristic
$\gamma_{SFE}$ copper	75 erg/cm^3^	10	[61]
$\gamma_{SFE}$ brass	20 erg/cm^3^	10	[61]
$M^{ki}$	Matrix, 24 × 24 elements, dimensionless	11	Calculating using elastic dislocations model
$B^{ki}$	Matrix, 24 × 24 elements, dimensionless	11	Calculating using elastic dislocations model
$\tau_{c\_lat}^{\left(k\right)}$ copper	17.5 MPa	11	[21]
$\tau_{c\_lat}^{\left(k\right)}$ brass	22.3 MPa	11	[39]

**Table 2 materials-15-00760-t002:** Additional hardening $\Delta_{ad}$ vs. φ and $P_{m}$, for Cu and Brass polycrystals (125 elements), equivalent strain amplitude = 1.5%.

**Material**	**P_*m*_**	**φ, Degree**	**Δ_*ad*_, %**	**φ, Degree**	**Δ_*ad*_, %**
Copper	1	15	5.1	60	6.4
		30	5.2	75	7.0
		45	6.1	90	7.1
Brass	1	15	11.1	60	13.1
		30	12.3	75	16.1
		45	12.8	90	23.1
**Material**	**φ, Degree**	**P_*m*_**	**Δ_*ad*_, %**	**P_*m*_**	**Δ_*ad*_, %**
Copper	33	0.25	1.1	0.75	4.1
		0.5	1.2	1	5.2
Brass	33	0.25	6.1	0.75	11.1
		0.5	10.3	1	12.3

## Data Availability

The data presented in this study are available on request from the corresponding author. The data are not publicly available due to privacy.

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
