# Peer review of "The Three-Level Elastoviscoplastic Model and Its Application to Describing Complex Cyclic Loading of Materials with Different Stacking Fault Energies"

_materials, 2022, doi:10.3390/ma15030760_

Round 1

Reviewer 1 Report

This submission cannot be accerpted for publication because the presentation is very bad.

Many sentences cannot be understood.

The equations are hard to comprhend.

Therefore the reader cannot judge whether the results are sound.

Author Response

The authors are grateful to the Reviewer for the labor of reading the manuscript of the article. Unfortunately, the Reviewer did not mark specific fragments of the manuscript that require cardinal revision, and therefore the authors worked through the entire text of the presented article, taking into account the broad comments made. We hope that in the new edition the content of the article has become clearer and more accessible to the reader.

Reviewer 2 Report

Manuscript by P. Trusov and D. Gribov entitled “The Three-Level Elastoviscoplastic Model and Its Application To Describing Complex Cyclic Loading of Materials With Different Stacking Fault Energies” describes a three-level mathematical model intended for the description of complex loads with an emphasis on complex cyclic deformation. In the Introduction authors highlight the problems encountered in the design and manufacturing of metals and alloys for special applications which require the development of constitutive models that can depict changes in the physical and mechanical characteristics of such materials upon thermomechanical treatment. The introductory part of the manuscript is well supported by references. In the following sections the mathematical model is described as well as its application for describing the response of a fcc polycrystalline macrosample, and in particular for the description of its cyclic deformations. The authors show that the results of numerical experiments obtained using the proposed model are in acceptable qualitative agreement with the known experimental data. In my opinion, the presented conclusions are adequately drawn and supported. 

Author Response

The authors are grateful to the Reviewer for the work done and the benevolent assessment of the submitted manuscript of the article.

Reviewer 3 Report

The manuscript titled "The Three-Level Elastoviscoplastic Model and Its Application 2 To Describing Complex Cyclic Loading of Materials With 3 Different Stacking Fault Energies" aims to present a three-level mathematical model of the evolution dislocations density and barriers, for describing complex cyclic deformation. The language is good to go. Since this paper is a theoretical study, regarding the contents, some improvements should be made before making further decision.

  1. The authors described the three-level model in detail using many formulas in section 2. However, the architecture or structure of the three-level model is not very clean to the readers. The implementation was described from line 278 to line 297 in text. Can the authors add some illustration to more clearly clarify the relation, function of the three-level model, maybe add a flow chart to present the implementation algorithm of this model?
  2. In section 3, the results and application section, the authors mainly use three stress-strain figures to present their findings and results. However, the details in terms of material, parameters in formulas, and modeling process, are not much provided. The readers might be confused about how these curves are produced, and what relation these curves are linked to the proposed formulas in section 2.
  3.  The authors stated that the three-level model is based on explicit description of the evolution of dislocation density. However, no results regarding the dislocation density are provided in section 3. Similar with question 2, can the authors provide more contour results of other quantities, such as dislocation density. In my opinion, the simple stress-strain curves don't convey much meanings or reflect the values of the three-level model.
  4.  In sum, the authors should present the three-level model more clearly if any flow-chart or diagram can be used. Besides, more results regarding other quantities using contour if applicable, should be supplemented to demonstrate the values of the model.

Author Response

The authors are grateful to thy Reviewer for the labor of reading the manuscript of the article, the authors tried to answer the questions asked:

  1. Q: The authors described the three-level model in detail using many formulas in section 2. However, the architecture or structure of the three-level model is not very clean to the readers. The implementation was described from line 278 to line 297 in text. Can the authors add some illustration to more clearly clarify the relation, function of the three-level model, maybe add a flow chart to present the implementation algorithm of this model?

A: The authors are grateful to the Reviewer for this constructive remark. When revising the manuscript, the authors took into account this remark and introduced the flowchart of the algorithm for implementing the proposed model.

  1. Q: In section 3, the results and application section, the authors mainly use three stress-strain figures to present their findings and results. However, the details in terms of material, parameters in formulas, and modeling process, are not much provided. The readers might be confused about how these curves are produced, and what relation these curves are linked to the proposed formulas in section 2.

A: The authors are very grateful to the Reviewer for the above remark, a table with the parameters used in the model was added at the beginning of the third section.

  1. Q: The authors stated that the three-level model is based on explicit description of the evolution of dislocation density. However, no results regarding the dislocation density are provided in section 3. Similar with question 2, can the authors provide more contour results of other quantities, such as dislocation density. In my opinion, the simple stress-strain curves don't convey much meanings or reflect the values of the three-level model.

A: The authors are very grateful to the Reviewer for the above remark, in section three plots of dislocation and barrier densities were added for a representative volume of brass and copper.

  1. Q: In sum, the authors should present the three-level model more clearly if any flow-chart or diagram can be used. Besides, more results regarding other quantities using contour if applicable, should be supplemented to demonstrate the values of the model.

A: The authors are very grateful to the Reviewer for the above remark. Two figures were added to the work: 1) scheme for conceptual setting (figure 1), 2) Schematic flow chart of model’s algorithm (figure 2). We hope that the structure of the model has become clearer.

We hope that in the new edition the content of the article has become clearer and more accessible to the reader. Please see the attachment.

Reviewer 4 Report

Comments on “The Three-Level Elastoviscoplastic Model and Its Application 2 To Describing Complex Cyclic Loading of Materials With 3 Different Stacking Fault Energies”:

The paper developed a three-level elastoviscoplastic model considering the effect of SFE to describe complex cyclic loading. The model covers the material behaviors in both macroscale and mesoscale, including dislocation slip and the interaction with defects. This model aims to develope the physical model for cyclic loading, especially for those materials with SFE. But there are some issues that need further explanation:

  1. Usually positive and negative dislocations are not distinguihed from each other for simplicity. What's the spefific benifit brought by considering dislocations with both signs considered explicitly? Is it disigned for cyclic loading? If yes, it would be help to explain and discuss in more detail.
  2. How does the SFE influence the model and the material behavior under cyclic loading?
  3. What is the main factor in this three-level model that is responsible for the advancement for modeling of complex cyclic loading? And can you provide the difference between this model and other CPFE models which also were disigned for cyclic loading or have the capabilty to model it?
  4. In section 3, the geometries of macrosample and boundary conditions should be provided, and a sketch map is desirable.
  5. In Page 7, line 301: “the law of uniform orientation distribution of crystal lattices” is not clear, may be random orientation distribution?
  6. In Eq. 5, it could be deduced that V+ and V- are inqual in maganitude all the way, but the text says it can be different.  Please explain it.

Author Response

The authors are grateful to thy Reviewer for the labor of reading the manuscript of the article, the authors tried to answer the questions asked:

  1. Q: Usually positive and negative dislocations are not distinguihed from each other for simplicity. What's the spefific benifit brought by considering dislocations with both signs considered explicitly? Is it disigned for cyclic loading? If yes, it would be help to explain and discuss in more detail.

A: The authors are very grateful to the Reviewer for the above remark. The proposed model uses the separation of dislocations located in the same plane and having the same (up to direction) Burgers vectors into positive and negative ones. This separation is required to describe several mechanisms of interaction of dislocations with each other (annihilation of dislocations, reaction of dislocations with the formation of barriers of a dislocation nature) and with one-sided barriers (for example, grain boundaries). In this case, the densities and velocities of motion of positive and negative dislocations of each slip system can differ.

  1. Q: How does the SFE influence the model and the material behavior under cyclic loading?

A: We are very grateful to the Reviewer for the above remark. The value of the SFE under any loading affects the critical stresses and hardening; for example, in materials with a low SFE, dislocations split, which makes it difficult for their non-conservative movement (cross-slips) and overcoming barriers. Split dislocations tend to form rigid barriers, especially under complex cyclic loading. In presented model the rate of formation of barriers on split dislocations directly depends on the SFE value (eq.9).

  1. Q: What is the main factor in this three-level model that is responsible for the advancement for modeling of complex cyclic loading? And can you provide the difference between this model and other CPFE models which also were designed for cyclic loading or have the capability to model it?

A: The authors are very grateful to the Reviewer for the above remark. The main advancement in our model is direct modeling of dislocations evolution, including reproduction of dislocations, annihilation; special attention is paid to the influence of the value of the stacking fault energy on the formation of dislocation barriers (Lomer – Cottrell, Hirth), which cause significant strengthening along slip systems. In the revised version of the article, an analysis of a number of works (sources 50-55) on this topic has been added, the features of the proposed model are noted.

  1. Q: In section 3, the geometries of macrosample and boundary conditions should be provided, and a sketch map is desirable.

A: We are very grateful to the Reviewer for the above remark. The term macro sample was removed from the beginning of the third section, so as not to mislead the reader. At this stage, we consider the deformation of only a representative volume of the material. In this case, the boundary value problem is not posed and not solved; for the considered representative volume, the homogeneity of the prescribed total gradients of the displacement velocities is assumed.

  1. Q: In Page 7, line 301: “the law of uniform orientation distribution of crystal lattices” is not clear, may be random orientation distribution?

A: The authors are very grateful to the Reviewer, we took into account your remark, corresponding corrections were made to the specified fragment of the text.

  1. Q: In Eq. 5, it could be deduced that V+ and V- are in equal in magnitude all the way, but the text says it can be different.  Please explain it.

A: Thanks for the noticed incorrectness, we add indicated signs for critical stresses and mean free path in equation 5.

We hope that in the new edition the content of the article has become clearer and more accessible to the reader. Please see the attachment.

Round 2

Reviewer 3 Report

The authors have carefully addressed my questions. In my opinion, the manuscript can be accepted in current form.